# Marketing Strategy, Serving Size, and Nutrition Information of Popular Children’s Food Packages in Taiwan

**DOI:** 10.3390/nu11010174

**Published:** 2019-01-15

**Authors:** Mei Chun Chen, Yi-Wen Chien, Hui-Ting Yang, Yi Chun Chen

**Affiliations:** 1School of Nutrition and Health Sciences, Taipei Medical University, 220 Taipei, Taiwan; gshejenny@gmail.com (M.C.C.); ychien@tmu.edu.tw (Y.-W.C.); 2Graduate Institute of Metabolism and Obesity Sciences, College of Nutrition, Taipei Medical University, Taipei 110, Taiwan; 3Department of Food Safety, College of Nutrition, Taipei Medical University, Taipei 110, Taiwan; d301091009@tmu.edu.tw

**Keywords:** child-targeted marketing, serving size, health and nutritional marketing, nutrition labeling, nutritional quality

## Abstract

A content analysis was used to investigate the marketing strategies, serving size, and nutrition quality in Taiwan popular children’s snacks and drinks. A total of 361 snacks and 246 drinks were collected. It was found that 38.6% of snacks and 25.3% of drinks were child-targeted (CT) foods, and 78.1% and 85.4% of the snacks and drinks had health and nutrition marketing (HNM). Serving size was significantly positively correlated to calories among different food categories in this study. Only the CT breads, ready-to-eat cereals, and fruit/vegetable juice had smaller serving sizes than did the corresponding non-CT products. These CT products had significantly fewer calories than did the corresponding non-CT products. Approximately 30% of snacks and 18% of drinks had both CT and HNM. Moreover, 82.7% of CT snacks and 100.0% of CT drinks with HNM were high in sugar. About 95% of foods with no added sugar claim were high sugar. CT foods are not necessarily healthier than non-CT foods, even the CT food with HNM. Health professionals should help parents assess the nutrition quality of the popular children’s foods. Further research was needed to investigate the effect of these marketing strategies and serving size on children’s food consumption.

## 1. Introduction

Obesity and being overweight increases the risk of many health problems, including diabetes, heart disease, and certain cancers [1,2,3]. Compared with the 2015 data of the World Obesity Federation, the overweight rates of children in Taiwan were the highest in Asia [4]. Maintaining a healthy diet is essential for preventing obesity and becoming overweight. Thus, the World Health Organization (WHO) recommends that children should consume foods with low saturated fat, trans fat, sugar, and sodium levels and that the marketing of foods with high levels of these components should be reduced [5]. However, a study in Brazil found that unhealthy products were often marketed using child-targeted (CT) strategies, such as cartoons and images of animals and celebrities on food packaging [6]. A supermarket survey in Canada also found that 91% of CT-food were high in sugar, fat, or sodium [7]. However, there is no study about marketing situation or nutrition content in the CT food in Asia.

In addition to the nutrition content of the food, the serving size provided to the children should also be noted. When children are served food in large serving sizes, this may lead to increased energy intake [8,9]. The increase of portion trends coincided with the increasing prevalence of obesity in both the United States and Europe. It has been speculated that they are causally connected [10,11,12]. Although these observational data cannot establish causality, they highlight the complexity of establishing a direct causal link between portion size and obesity, given that energy intakes are a function of not only the portion size of food, but also its energy density and the frequency of food and beverage consumption, among other factors.

Parents are often attracted to foods with health and nutrition information, such as composition, nutrition, or health claims. In the United States, Colby et al. noticed that although most nutrition and health marketing studies were focused on the impact of television advertising, nutrition marketing used on food labels may also influence consumer’s consumption patterns [13]. An online survey in Australia found that parents who perceived the products with nutritional claims, such as source of fibers, source of calcium, reduced fat, on food packaging as nutritious were more likely to purchase such products [14].

Eating snacks is growing in importance as an eating occasion for children across the world [15]. Wang et al. indicated that providing healthy snacks could increase energy and nutrition intake for children and improve the nutritional quality of children’s diets [16]. In the United States, childhood snacking has increased to an average of three snacks or beverages per day, representing 27% higher daily energy intake than is recommended [17]. However, a supermarket study in the United States found that 49% of packaged products (e.g., baby foods, snacks, dairy and egg products, and breakfast cereals) used nutrition marketing, such as structure/function claims, nutrient content claims, and health claims, and that 48% of them were high in sugar, saturated fat, or sodium [13].

Currently, many countries have mandatory legislation on nutrition labeling to help people to make healthy choices about foods. A total of 35 countries require or recommend that nutrition information is presented as per serving size; however, the definition of one serving is different [18]. The definition of serving size is the average amount typically consumed in one occasion in Australia, New Zealand, Canada, the USA, and the European countries, and other countries, such as Argentina, Brazil, Paraguay, and Uruguay, have stablished recommended serving sizes for foods [15]. In 2018, the Taiwan government made it mandatory for food manufacturers to present the nutrient content of their products per serving size; however, the serving size among similar food is not necessarily the same and the reference of a serving size is established by consumers’ dietary pattern and common food package capacity and weight [19]. Studies on the serving sizes for children have focused on children’s energy intake [8,20,21] or assessment nutrition quality of CT food [6,7]. Therefore, this study investigated the nutritional content of various serving sizes of popular children’s snacks and drinks and examined the marketing strategies and nutritional quality of these children’s foods. The current results may serve as a guide for related education and policies of CT food.

## 2. Materials and Methods

### 2.1. Data Collection

Content analysis was used to investigate the associations among the marketing strategies, nutritional quality, and serving sizes of popular children’s foods. We included commercial snacks, including (1) cookies (biscuits and crackers), (2) breads, (3) ready-to-eat cereals, and (4) puddings or jellies, as well as drinks, including (1) fruit or vegetable juices, (2) flavored milks: processed milk products containing at least 50% raw, fresh, or sterilized milk mixed with flavors; (3) fermented milks: milk that was fermented with lactic acid bacteria; (4) soy and rice milks: unflavored or flavored; and (5) milk teas: black milk tea, oolong milk tea and strawberry milk teas. These products were usually used as Taiwanese children’s snacks and drinks that are provided between regular meals [22]. Candies, potato chips, and soda, which contain a lot of calories from sugar or fat with little dietary fiber, protein, vitamins or minerals, were excluded [7].

Data was collected between November 2017 and March 2018 in chain supermarkets (i.e., PXMart, Carrefour, and Simple Mart) and convenience stores (i.e., 7-Eleven and Family Mart) in Taipei. These supermarket and convenience stores have a high market share in Taiwan (the top three for supermarket and top two for convenience stores). The researchers visited at least two different branches on each chain supermarket/convenience stores. A total of 11 stores were visited and 9 categories of products were collected. Food packaging information of each product was obtained by viewing the products in stores or from the manufacturers’ websites, in accordance with methods used in a previous study in U.K. [23]. This information was photocopied for reference and stored for subsequent analysis. This research was conducted with no human subject and all collected data is publicly available, therefore no approval from an ethical commission was required.

### 2.2. Coding Procedure and Coded Content

During the coding procedure, common themes were identified and recorded. The first author developed a coding form from the gathered information and then used the initial coding form to instruct the other two researchers, who confirmed whether the coding was correct. The three researchers were subsequently responsible for coding all photographed information. The first author was consulted for difficult coding decisions, such as defining health symbols. A pretest verified that the coding form included the distinct categories required to classify the contents of the food packages; however, several new categories, such as composition claim, nutrition claim, and health claim, were added if required during the coding process. This process also verified whether all the researchers understood and used these categories in the same way.

Table 1 presents the coded component of food packaging information including general product information, product category, CT strategy, healthy and nutrition marketing (HNM) strategy, nutritional information, and nutritional quality. The target age was defined as marketing on products being perceived to be targeting the young child (1–8 years), preteen (9–12 years), or both. A snack or drink was considered to have a CT strategy if the packaging had at least one of the following: (1) an attractive picture or bright colors, which could appeal to children; (2) images of promotional characters, such as brand-specific characters, cartoon or movie characters, animals, and popular celebrities; (3) reference to children; (4) games; (5) specified age ranges; and (6) free gifts [6,7].

### 2.3. Nutritional Information and HNM Strategies

The nutritional information per serving size listed on the food packaging, including calories and the carbohydrates, proteins, total fat, saturated fatty acid (SFA), sugar, and sodium content, was coded. Snacks and drinks were considered of poor nutritional quality if they met at least one of the following criteria: (1) High sugar: according to WHO [24] and Taiwan Ministry of Health and Welfare [25] recommendations, children should derive less than 10% of their daily calories from sugar. Therefore, foods with 10% or more calories derived from sugar were defined as having high sugar content. (2) High fat content: the Taiwan Ministry of Health and Welfare [25] recommended children should derive less than 30% of their daily calories from fat. Therefore, foods with 30% or more calories derived from fat were defined as having high fat content. (3) High SFA content: both WHO [26] and Taiwan Ministry of Health and Welfare [25] recommended children should derive less than 10% of their daily calories from SFA. Therefore, foods with 10% or more calories derived from SFA were defined as having high SFA content. (4) High sodium content: there is no recommended about sodium content per serving in Taiwan. A previous study in Canada noticed that for 9–50-year-olds, it would be 200 and 400 mg sodium per serving [27]. Therefore, foods with 200 mg sodium per serving were defined as having high sodium content.

The HNM strategies were coded and categorized into (1) healthy symbols, such as fresh fruits or vegetables or a natural scenery [28]; (2) composition claims, namely overarching representations of food ingredients or food qualities (e.g., “organic”) and the exclusion of particular ingredients (e.g., “no preservatives” or “gluten-free”) [29]; (3) nutrition claims, namely any representation stating or suggesting that a food has or excludes a particular nutrient (e.g., “no trans-fat” or “contains fiber”) [30,31]; (4) health claims, namely any representation stating or suggesting that a relationship between the food and a health benefit (e.g., dietary fiber improves digestion) or claiming that consuming the food (or its constituents) influenced the normal functions or biological activities of the body in the context of the wider diet (e.g., contributing toward balanced nutrition) [30]. Only some health claims here were approved as health maintenance claims, since in Taiwan, there is no regulation regarding health claims. According to “Health Food Control Act”, health maintenance claims are allowed for “health Food” whose health maintenance effects are examined and approved by the Taiwan Ministry of Health and Welfare [32]. (5) Product names symbolizing the healthiness, e.g., “nutritious”, “energetic”, and “nature” [28].

Sugar- and fat-related claims were also coded. “Sugar free”, “reduced sugar”, “no sugar added”, and “no artificial sweeteners” were considered low sugar–related claims. “Fat free”, “low fat”, “no trans-fat”, “no cholesterol”, and “not fried” were considered low fat–related claims.

### 2.4. Interrater Reliability and Data Analysis

Kappa statistic tests were conducted to test the interrater reliability of the two coders (kappa coefficient strength was defined as 0.41–0.60, moderate agreement; 0.61–0.80, substantial agreement; and 0.81–1.00, almost perfect agreement [33]). Intercoder reliability was pilot-tested among three researchers by using the original content analysis form. The final kappa analysis was calculated from the final coding of 50% of food packages that were coded by three researchers; the kappa statistics were higher than 0.80 and were thus deemed acceptable.

All statistical analyses were conducted using SPSS version 19.0 (SPSS Inc., an IBM Company, Chicago, IL, USA). The results are presented as frequencies (*n*), percentages (%), or median (interquartile range (IQR)). The Kolmogorov–Smirnov test was used to examine the normal distribution of all continuous variables. Serving size, calories, and nutrition content were not normally distributed. Spearman’s rank correlation analysis was used to measure the association between serving size and calories value among different foods. Kruskal–Wallis H test and post hoc Dunn test were used to compare the serving size and calories among different foods. The Mann–Whitney *U* test was used to compare the serving sizes and nutritional information between CT and non-CT products in each category. Pearson’s chi-square test or Fisher’s exact test was used to compare the levels of poor nutritional quality among the different foods, as appropriate. For all analyses, *p* ≤ 0.05 was considered significant.

## 3. Results

### 3.1. Marketing Strategy Distribution of Snacks and Drinks

As depicted in Figure 1, 361 snacks and 246 drinks were collected. Furthermore, 38.8% of the snacks and 25.6% of the drinks were CT foods. There were 78.1% and 85.4% of the snacks and drinks with HNM, respectively.

### 3.2. CT and HNM

Figure 2 presents the distribution of the CT strategies. Cute pictures or bright colors and promotional characters were used respectively by 60.0% and 47.1% of the CT snacks and by 63.5% and 29.7% of the CT drinks, respectively. Moreover, 12.9% of the CT snacks had a reference to children and 1%–5% of the CT foods used games, specified age ranges, or offered free gifts.

Table 2 shows the distribution of the identified HNM strategies. Regarding the HNM strategies of the snacks and drinks, the two most frequently used HNM strategies were health symbol and composition claims. Moreover, the top three composition claims were “no added colors” (*n* = 119), “no added preservatives” (*n* = 99), and “no added flavors” (*n* = 67). The results also revealed that the CT snacks frequently used composition claims (49.3%), whereas the CT drinks commonly used images related to health (50.8%).

### 3.3. Serving Sizes and Nutritional Information of Products

Table 3 lists the association of serving size, calories, and nutrition content among different foods. The range was 19.3–190.0 g for pudding/jelly and 125.0–650 mL for fruit/vegetable juice. The serving size of pudding/jelly was significantly larger than the other snacks. The serving size of milk tea was significantly larger than fermented milk and fruit/vegetable juice drinks. Serving size positively correlated to calories in all food categories, but media association was found in large serving size food categories. A strong positive association was found between serving size and calories for most snacks and drinks (*r* > 0.80). A medium association was found in pudding/jelly (*r* = 0.46) which energy density was 1.0, and milk tea (*r* = 0.58), whose energy density was 0.53.

Table 4 compares the serving sizes and nutritional information between the CT and non-CT foods. Among food categories, common CT foods were cookies, ready-to-eat cereals, and flavored milks (48.3%, 57.4%, and 36.7%, respectively). No significant differences were observed between the serving sizes or the nutritional information of the CT and non-CT cookies. Both the CT breads and ready-to-eat cereals had smaller serving sizes than did the corresponding non-CT products. Moreover, the CT breads and ready-to-eat cereals had significantly fewer calories and lower carbohydrate, protein, and fat content than did the corresponding non-CT products. Although CT puddings or jellies had smaller serving sizes than did the corresponding non-CT products, no significant differences were observed between their nutritional information.

The CT fruit or vegetable juices and soy or rice milks had smaller serving sizes than did the corresponding non-CT products. The CT fruit or vegetable juices had significantly fewer calories and lower carbohydrate, sugar, and sodium content than did the corresponding non-CT products. Moreover, the CT soy or rice milks had less fat content than did the corresponding non-CT products. No significant differences were observed in the serving sizes or nutritional information of the CT and non-CT flavored milks and milk teas.

### 3.4. HNM Versus Nutritional Quality

Table 5 demonstrates the association between the adopted HNM strategies and the nutritional quality of the products. More than half of the snacks were high in sugar and fat (79.5%, 54.0%, respectively). Overall, 97.6% and 39.0% of the drinks were high in sugar and SFA, respectively. Moreover, 82.7% and 100% of the CT snacks and drinks that had HNM were high in sugar, respectively. Notably, 40–60% of the snacks with HNM were high in fat and SFA. About 20% of snacks and 2.4% of drinks were high sodium. Compare to the other foods, a lower high sodium content was found in the CT snacks and drinks with HNM.

### 3.5. Nutritional Quality of Products with Low Fat or Sugar-related Claims

Table 6 shows the nutritional quality of products with low fat–related and low sugar-related claims. Overall, 25.7% of all samples (*n* = 156) had a sugar- or fat-related claim, 91.7% of the products with a low sugar-related claim, and 85.7% of the products with a low fat-related claim were high sugar. Except “sugar free”, more than 90% of products with a low sugar-related claim were high sugar. More than 30% of products with “no cholesterol” or “not fried” were high fat.

## 4. Discussion

### 4.1. Marketing Strategy of Snacks and Drinks

This is the first study to compare serving sizes and nutritional information between CT and non-CT foods. In this study, about 40% of snack packaging and 25% of drink packaging contained marketing strategies targeted at children, particularly cookies, ready-to-eat cereals, and flavored milks. This finding is consistent with previous studies. A content analysis in Guatemala found 35.4% of 2334 food packaging contained marketing strategies targeted at children [34]. The other survey in U.K. supermarket found 40.8% of yoghurts and 29.7% of cereal bars were children-targeted [23]. The CT marketing strategies of snacks and drinks identified in the current study mainly comprised cute pictures or bright colors (approximately 60%) and promotional characters (30–50%), whereas 1–15% of food packaging contained references to children, contained games, specified age ranges, or offered free gifts. A Guatemalan study found that 92.5% of food packaging for products, such as savory snacks and soft drinks, used promotional characters [34]. A study in Brazil found that more than half of CT foods in large chain stores were sweets and drinks, and 35–50% of CT foods mainly used images aimed at children, including well known TV characters or brand characters [35]. A systematic review reported that CT marketing can directly influence children’s food preferences, purchasing choices, and food consumption [36]. Further research is needed to investigate the effect of CT marketing strategies on children’s eating behavior in Taiwan.

### 4.2. Serving Size, Calorie, and Nutrition Information

A great variation of serving size was observed among similar food products in this study, since in Taiwan, the serving size among similar foods are not necessarily the same. The reference of serving size is established by consumers’ dietary pattern and common food package capacity and weight [19]. The finding was consistent with the finding that was also observed among 1953 food products in Brazil [37], and among 1070 food in an Australian supermarket [38]. Serving size was significantly positively correlated with calories among different food categories in this study. Only a medium positive association was found for pudding/jelly and milk tea, which had large serving sizes. Compared to the other snacks, pudding/jelly has a high water content with low energy. A survey in the USA also found an inverse relation between serving size and energy density with those had a high water content and low energy density, such as soft drinks, juices, yogurts, and soups, had a large serving size (200–240 g) [39]. There was also a tendency to report smaller serving sizes for products with higher calorie densities among 10,487 food products in Canada [40]. Kliemann et al. indicated that varying serving sizes of products of poor nutrition quality was a marketing strategy for food companies [18]. Parents need to pay attention to compare the calorie value and serving size among different food categories.

In Taiwan, consumers’ dietary patterns were the reference of serving size. Children’s dietary patterns are different from adults’. However, no regulation for the different serving sizes of children’s foods have been implemented in Taiwan. Only the CT breads, ready-to-eat cereals, and fruit/vegetable juice had smaller serving sizes than did the corresponding non-CT products. Furthermore, these CT products had significantly fewer calories than did the corresponding non-CT products. Based on MyPlate, parents should prepare single servings and control the serving sizes that their children consume [41]. Providing large portions of energy-dense food to young children increases their daily energy intake [42]. In the United Kingdom and Ireland, childhood obesity prevention campaigns have identified serving size as a key issue in preventing child obesity and have emphasized the importance of offering appropriately sized food to children [43,44]. Ferrage et al. also suggested that because there is no association between the serving sizes of snacks and the portion selection among young children, reducing the size of snacks may be an effective strategy when providing snacks for young children [45]. In the future, it could be considered that stipulate different serving sizes for children food should be established.

### 4.3. Nutrition Quality versus Marketing Strategy

The current study found that half the CT snacks were high in fat, approximately 80% of the CT snacks and 100% of the CT drinks were high in sugar, and 9% of the CT snacks were high in sodium. A study found that “cereals for kids” had higher calories and sugar compared with other breakfast cereal categories [46]. A Canadian survey reported that approximately 90% of CT foods were high in fat, sugar, or sodium [7]. A study also found that 89% of CT products in Canadian supermarkets were high in fat, sugar, or sodium [47]. Compared to the other study, less high sodium CT foods were found in this study. This may due to potato chips or ready-to-eat meal being excluded in this study.

In the current study, 78.1% of the snacks and 85.4% of the drinks had HNM. Another study in Taiwan found that nearly 90% of commercial infant food had HNM [48]. A Guatemalan study also found that 41% of snack foods had nutritional and health claims [34]. Moreover, approximately 30% of the snacks (*n* = 110) and 18% of the drinks (*n* = 46) in the current study contained both CT and HNM, meaning that these products target both children and parents. However, 82.7% of the CT snacks with HNM and 100.0% of the CT drinks with HNM were high in sugar. These results are similar to those reported in related studies that CT food often had high sugar or high fat problem [13,28,47]. A study on Canadian supermarket products reported that 89% of CT products had poor nutritional quality and were high in sugar, fat, or sodium, and that 60% of these products had HNM (e.g., nutritional claims or health claims) on the packaging [47]. A U.S. study also found that 71% of CT products had nutritional marketing, and of these, 59% were high in SFA, sodium, or sugar, with more than 50% being high in sugar [13]. Moreover, a U.S. study suggested that products with nutritional and CT marketing were higher in sugar compared with products without such marketing [28]. Thus, parents need to be aware of the associations between food package marketing strategies and the nutritional quality of the products. Especially, they need to know that products that generally target children are typically less healthy.

Overall, the CT foods in the current study were not healthier than the non-CT foods, even in the case of those with HNM. An Australian study found that parents were misled by the nutritional claims on CT foods that had poor nutrition; the parents believed that such products were more nutritious than the same foods without nutritional claims [14]. Therefore, the government should regulate the food claims or health-related marketing of CT foods. Reducing the visual appeal of the packaging of unhealthy foods may also be an effective strategy.

A U.S. study found that some parents were concerned their children’s intake of high-fat and sugary foods [49]. A study in New Zealand also reported that parents were concerned about their children’s sugar and fat intake [50]. Products with low or no fat-related claims and those with low or no sugar-related claims may be particularly appealing to these parents. However, we found that 85.7% of the products with low fat-related claims were high in sugar and that more than 90% of products with no added sugar or with no added artificial sweeteners, were high in sugar. Similarly, a Canadian study examined 3048 prepackaged foods with sugar-related claims (such as “sugar free”, “no added sugar”, and “reduced sugar”) and found that almost 50% were high in sugar [51]. Moreover, a study found that 28% of consumers misunderstood the meaning of the “no added sugar” claim, believing that products with such a claim contained no sugar [52]. A Taiwanese survey also found that 40% of mothers misunderstood “no added sugar” to be “sugar free” or “less sugar”, and they considered products with such a claim as healthy, but the mothers with higher sugar-related knowledge had a lower intention to purchase products with no added sugar [53]. Dixon et al. noticed that parents who read nutrition labels when purchasing children’s food are less likely to make unhealthy food choices than those who do not [14]. To reduce the chances of parents and caregivers being misled by HNM, the government should provide nutrition education to parents and caregivers and enhance the regulation of marketing strategy on children popular food packaging.

This study has some limitations. First, the sample might not include all popular commercial children’s snacks and drinks in Taiwan. However, we did our best to collect data from 11 branches which belong to the top three high market share of Taiwan supermarket and the top two for convenience stores in Taiwan. Second, we did not use instrumental analysis to verify the nutritional content of the products; thus, we could not comprehend determine and compare the nutritional content of all the sampled commercial snacks and drinks.

## 5. Conclusions

Popular children’s snacks and drinks that try to engage both young children and their parents are not necessarily healthy. Parents need to pay attention and compare the calorie value and serving size among different food categories. Cookies that had a higher energy density than the other snacks had a great variety of serving size. Pudding/jelly that had a high water content had low energy density. Health professionals should help parents interpret and assess the nutritional information on such products. Government policies must be directed toward the HNM used by food manufacturers, particularly fat- and sugar-related claims, such as “no added sugar” or “no added artificial sweeteners”, because they may be misleading.

## Figures and Tables

**Figure 1 nutrients-11-00174-f001:**
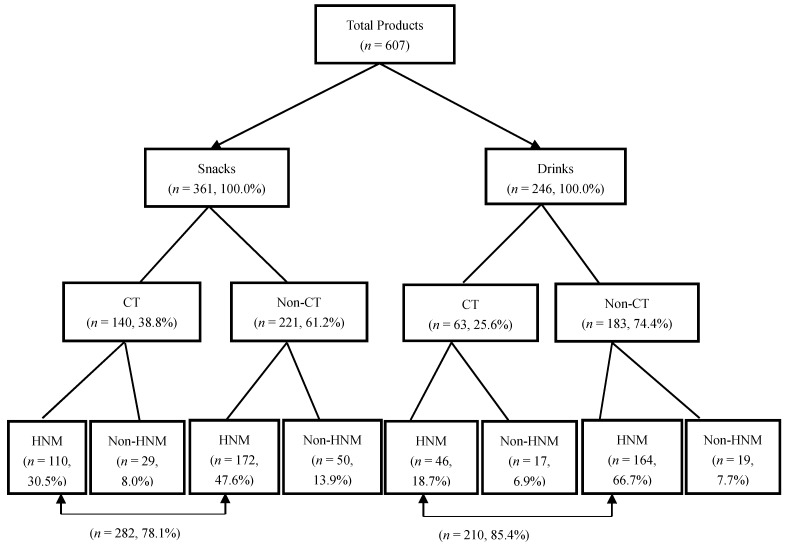
Sample distribution. CT: child-targeted; HNM: health and nutrition marketing.

**Figure 2 nutrients-11-00174-f002:**
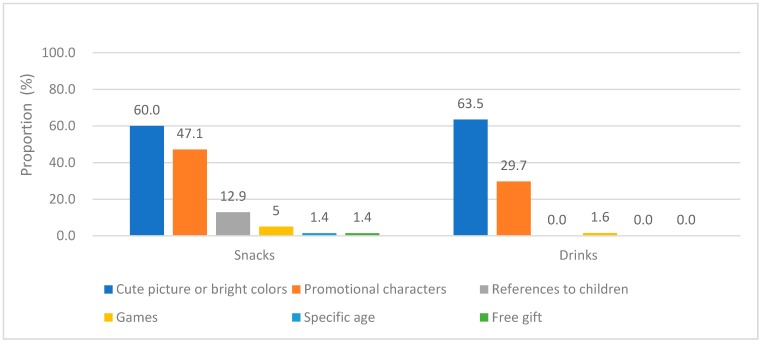
CT strategies in snacks (*n* = 361) and drinks (*n* = 246).

**Table 1 nutrients-11-00174-t001:** Coded component of food packaging information.

Coding Content	Coding Standards
General	Product name
Information	Brand name
Product classification	Snacks or drinks
Product category	Cookies, bread, ready-to-eat cereal, pudding/jelly, fruit/vegetable juice, flavored milk, fermented milk, soymilk/rice milk, milk tea
Child-targeted strategy	Cute picture or bright colors, promotional characters, references to children, games, specified age ranges or free gifts
Nutrition facts	Calorie, carbohydrate, protein, total fat, saturated fat (SFA), sugar, and sodium content
Nutritional quality	High fat ^1^, high SFA ^2^, high sugar ^3^, high sodium ^4^
Health and nutrition marketing strategy	Health symbol, composition claim, nutrition claims, health claim, product names symbolizing the healthiness
Health symbol	Fresh fruits or vegetables picture, a natural scenery
Composition claim	No added coloring agent, no added preservative, no added flavoring agent, no artificial sweeteners, non GMO (Genetically modified organism) food, not fried, milk content above 50%, no added food additive, organic food, gluten-free, nature source, fresh yeast, no coffee cream, whole grain, no allergens, non-hydrogenation, additives from plants, contains plant oil, no added glutamate flavoring, contains fresh milk, contains health oligosaccharides, modified starch free, contains nuts, contains vegetables, contains fruits, no food additives, nature fruit and vegetable juice, plant milk, no added sugar, no added salt
Nutrition claims	No trans-fat, no cholesterol, no fat, low fat, sugar free, contains probiotics and/or prebiotic, contains dietary fiber, contains calcium, contains several vitamins, contains iron, contains oligosaccharides, contains protein or amino acid, low sodium, contains vitamin C, contains vitamin E, contains variety mineral, reduced/low calories, reduced sugar, contains vitamin B1, contains vitamin B2, contains folate, contains ω-3, contains vitamin A, contains vitamin D
Health claims	Good for digestion and absorption, good for constitution, good for health, good for enteric flora, nutritionally balanced, good for growth, improve constipation, vitality, good for iron absorption, good for teeth and bone
Product names symbolizing the healthiness	Nature, health, nutrition, energetic
Sugar-related claims	Sugar free, reduced sugar, no added sugar and no added artificial sweeteners
Fat-related claims	Fat free, low fat, no cholesterol, no trans-fat, not fried

^1^ High fat: foods whereby fat content represented >30% of product calories. ^2^ High SFA: foods whereby SFA content represented >10% of product calories. ^3^ High sugar: foods whereby sugar content represented >10% of product calories. ^4^ High sodium: foods whereby the sodium content represented >200 mg sodium per serving.

**Table 2 nutrients-11-00174-t002:** Marketing strategies used on packaged snacks and drinks ^1^.

	Snacks			Drinks		
	Total	CT	Non-CT	Total	CT	Non-CT
HNM strategy ^2^	361 (100.0)	140 (100.0)	221 (100.0)	246 (100.0)	63 (100.0)	183 (100.0)
Health symbol	167 (46.3)	58 (41.4)	109 (49.3)	158 (64.2)	32 (50.8)	126 (68.9)
Composition claim	163 (45.2)	69 (49.3)	94 (42.5)	94 (38.2)	21 (33.3)	73 (39.9)
Nutrition claim	71 (19.7)	42 (30.0)	29 (13.1)	83 (33.7)	16 (25.4)	67 (36.6)
Health claim	8 (2.2)	8 (5.7)	0 (0.0)	22 (8.9)	5 (7.9)	17 (9.3)
Product names symbolizing the healthiness	14 (3.9)	5 (3.6)	9 (4.1)	8 (3.3)	1 (1.6)	7 (3.8)

^1^ Data are presented as number (percentage). ^2^ One package can have more than one marketing strategy. CT: child-targeted; HNM: health and nutrition marketing.

**Table 3 nutrients-11-00174-t003:** The association of serving size and calorie among different foods ^1^.

	*N*	Serving Size Range	Serving Size (g or mL) ^2^	Calories (kcal) ^2^	Calories Density ^2^	Serving Size vs. Calories (*r*) ^3^
Snacks						
Cookies	176	4.0–88.0	26.0 (14.8) ^a^	131.7 (71.1) ^a^	4.99 (0.49) ^c^	0.97 **
Bread	80	11.0–180.0	101.0 (33.0) ^c^	319.0 (163.1) ^b^	3.43 (1.00) ^b^	0.87 **
Ready-to-eat cereal	68	20.0–50.0	35.0 (15.0) ^b^	132.8 (48.6) ^a^	3.82 (0.26) ^b^	0.97 **
Pudding/jelly	37	19.3–190.0	130.0 (65.0) ^c^	112.4 (57.0) ^a^	1.00 (0.37) ^a^	0.36 *
Drinks						
Fruit/vegetable juice	66	125.0–650.0	250.0 (100.0) ^b^	107.5 (54.3) ^a^	0.43 (0.09) ^a^	0.79 **
Flavored milk	60	125.0–440.0	285.0 (114.5) ^bc^	177.6 (68.8) ^c^	0.64 (0.09) ^b^	0.87 **
Fermented milk	44	100.0–359.0	213.0 (113.4) ^a^	132.6 (82.2) ^ab^	0.65 (0.16) ^b^	0.87 **
Soymilk/rice milk	41	190.0–450.0	250.0 (187.5) ^bc^	133.3 (81.9) ^b^	0.52 (0.18) ^c^	0.78 **
Milk tea	35	236.5–400.0	315.3 (132.5) ^c^	156.0 (66.5) ^bc^	0.53 (0.19) ^c^	0.60 **

^1^ Data are presented as median (IQR) nutritional content per serving. ^2^ Using a Kruskal–Wallis H test. ^3^ Using a Spearman’s rank correlation analysis. * *p* ≤ 0.05, ** *p* ≤ 0.001. Values in a column not sharing the same superscript letter (a–c) differed significantly, *p* < 0.05.

**Table 4 nutrients-11-00174-t004:** Serving sizes and nutritional information of CT and non-CT foods ^1^.

	*N* (%)	Serving Size	Calorie (kcal)	Carbohydrate (g)	Protein (g)	Fat (g)	Saturated Fat (g)	Sugar (g)	Sodium (mg)
		Median (IQR)							
Cookies									
CT ^2^	85 (48.3)	26.0 (16.0)	131.3 (83.1)	18.6 (8.5)	1.6 (1.1)	5.9 (6.0)	3.0 (3.2)	5.4 (7.1)	50.0 (131.4)
Non-CT	91 (51.7)	25.4 (13.4)	131.8 (66.5)	16.5 (8.0)	1.8 (1.5)	5.8 (3.1)	2.9 (1.6)	4.7 (7.4)	78.0 (88.3)
*p*-value		0.362	0.993	0.004 *	0.279	0.601	0.537	0.086	0.483
Bread									
CT ^2^	6 (7.5)	55.0 (48.8)	192.2 (157.5)	31.4 (28.7)	4.1 (4.0)	4.2 (4.9)	1.5 (1.3)	10.1 (11.6)	87.1 (58.5)
Non-CT	74 (92.5)	102.0 (30.0)	347.1 (152.8)	49.6 (16.0)	8.3 (3.5)	12.9 (13.3)	5.0 (5.8)	15.1 (9.9)	289.4 (184.0)
*p*-value		0.001 *	0.001 *	0.004 *	0.002 *	0.009 *	0.003 *	0.132	<0.001 **
Ready-to-eat cereal									
CT ^2^	39 (57.4)	30.0 (15.0)	116.0 (54.3)	22.4 (13.3)	2.0 (1.0)	0.9 (1.9)	0.3 (0.4)	8.8 (4.8)	90.0 (81.0)
Non-CT	29 (42.6)	37.5 (14.0)	140.8 (50.8)	30.0 (10.2)	3.4 (2.1)	1.9 (2.5)	0.5 (0.6)	8.0 (4.1)	49.0 (105.8)
*p*-value		0.001 *	0.002 *	0.014 *	<0.001 **	0.014 *	0.269	0.170	0.076
Pudding/jelly									
CT ^2^	10 (27.0)	100.0 (34.5)	103.5 (77.1)	16.9 (13.7)	1.1 (2.3)	1.5 (3.5)	1.1 (2.6)	16.0 (20.1)	50.0 (66.2)
Non-CT	27 (73.0)	150.0 (55.0)	114.5 (50.9)	26.5 (8.9)	0.2 (4.0)	0.3 (2.6)	0.0 (0.8)	22.3 (8.3)	68.9 (58.4)
*p*-value		0.002 *	0.274	0.058	0.386	0.874	0.825	0.128	0.161
Fruit/vegetable juice									
CT ^2^	15 (22.7)	200.0 (75.0)	88.0 (40.5)	21.0 (10.6)	0.4 (1.0)	0.0 (0.0)	0.0 (0.0)	20.0 (8.9)	27.5 (24.9)
Non-CT	51 (77.3)	250.0 (105.0)	112.5 (57.5)	28.1 (16.0)	0.0 (0.6)	0.0 (0.0)	0.0 (0.0)	25.8 (13.4)	47.5 (42.0)
*p*-value		<0.001 **	0.004 *	0.003 *	0.033 *	0.340	1.000	0.005 *	0.010 *
Flavored milk									
CT ^2^	22 (36.7)	247.5 (100.0)	170.9 (66.1)	23.5 (11.6)	5.2 (2.7)	5.0 (1.9)	3.0 (1.5)	21.5 (10.7)	85.0 (62.5)
Non-CT	38 (63.3)	290.0 (100.3)	184.7 (80.4)	27.5 (14.1)	5.6 (2.4)	5.3 (3.2)	3.1 (2.1)	21.6 (15.0)	117.4 (81.7)
*p*-value		0.055	0.200	0.369	0.078	0.125	0.425	0.662	0.104
Fermented milk									
CT ^2^	11 (25.0)	170.0 (140.0)	108.8 (68.6)	25.2 (15.0)	2.2 (2.8)	0.0 (0.0)	0.0 (0.0)	23.8 (15.5)	42.5 (38.2)
Non-CT	33 (75.0)	242.9 (85.7)	140.0 (69.4)	25.9 (13.4)	6.2 (5.7)	1.2 (3.2)	0.8 (2.5)	20.7 (13.7)	103.0 (92.5)
*p*-value		0.052	0.104	0.946	0.009 *	0.017 *	0.019 *	0.440	0.029 *
Soymilk/rice milk									
CT ^2^	9 (22.0)	200.0 (50.0)	115.0 (35.2)	19.0 (9.3)	5.0 (1.4)	3.2 (0.4)	0.8 (0.1)	15.4 (9.9)	66.0 (77.5)
Non-CT	32 (78.0)	270.0 (153.8)	143.6 (99.9)	16.6 (20.6)	7.4 (7.2)	5.1 (3.7)	1.0 (0.8)	11.1 (18.3)	29.8 (61.9)
*p*-value		0.002 *	0.083	0.987	0.032 *	0.007 *	0.070	0.625	0.119
Milk tea									
CT ^2^	6 (17.1)	357.5 (114.3)	170.9 (36.8)	28.2 (8.3)	5.1 (3.2)	5.3 (2.5)	3.9 (1.6)	26.5 (5.2)	81.1 (37.5)
Non-CT	29 (82.9)	312.5 (141.3)	148.4 (74.3)	27.8 (8.6)	2.0 (3.2)	3.7 (4.3)	3.0 (3.0)	23.3 (9.6)	79.8 (42.0)
*p*-value		0.658	0.246	0.662	0.062	0.430	0.776	0.314	0.710

^1^ Data are presented as median (IQR) nutritional content per serving. * *p* ≤ 0.05, ** *p* < 0.001, Kolmogorov–Smirnov test and Mann–Whitney *U* test. ^2^ CT: child-targeted.

**Table 5 nutrients-11-00174-t005:** Nutritional quality of CT and non-CT foods with and without HNM ^1^.

	*N* (%)	CT		Non-CT	
		HNM	Non-HNM	HNM	Non-HNM
Total (Snacks)		110 (100.0)	29 (100.0)	172 (100.0)	50 (100.0)
High fat ^2^	195 (54.0)	48 (43.6)	20 (69.0)	91 (53.2)	36 (72.0)
High SFA ^3^	215 (59.6)	52 (47.3)	23 (79.3)	102 (59.3)	38 (76.0)
High sugar ^4^	287 (79.5)	91 (82.7)	20 (69.0)	135 (78.5)	41 (82.0)
High sodium ^5^	76 (21.1)	10 (9.0)	6 (20.7)	44 (25.7)	16 (32.0)
Total (Drinks)		46 (100.0)	17 (100.0)	164 (100.0)	19 (100.0)
High fat ^2^	30 (15.9)	5 (10.9)	3 (17.6)	29 (17.7)	2 (10.5)
High SFA ^3^	96 (39.0)	13 (28.3)	10 (58.8)	61 (37.2)	12 (63.2)
High sugar ^4^	240 (97.6)	46 (100.0)	17 (100.0)	158 (96.3)	19 (100.0)
High sodium ^5^	6 (2.4)	0 (0.0)	0 (0.0)	5 (3.0)	1 (5.9)

^1^ Data are presented as the number (percentage). ^2^ High fat: foods whereby fat content represented >30% of product calories. ^3^ High SFA: foods whereby SFA content represented >10% of product calories. ^4^ High sugar: foods whereby sugar content represented >10% of product calories. ^5^ High sodium: foods whereby the sodium content represented >200 mg sodium per serving. CT: child-targeted; HNM: health and nutrition marketing; SFA: saturated fatty acid.

**Table 6 nutrients-11-00174-t006:** Sugar- and fat-related claims and nutritional quality ^1^.

	Total	High Sugar ^2^	High Fat ^3^	High SFA ^4^
Low sugar related claim	72 (100.0)	66 (91.7)	7 (9.7)	8 (11.1)
Sugar free	4 (100.0)	0 (0.0)	2 (50.0)	1 (25.0)
Reduced sugar	3 (100.0)	3 (100.0)	1 (33.3)	0 (0.0)
No added sugar	24 (100.0)	23 (95.8)	3 (12.5)	7 (29.2)
No added artificial sweeteners	42 (100.0)	41 (97.6)	2 (4.8)	1 (2.4)
Low fat related claim	84 (100.0)	72 (85.7)	13 (15.5)	12 (14.3)
Fat free	9 (100.0)	9 (100.0)	0 (0.0)	0 (0.0)
Low fat	8 (100.0)	7 (87.5)	0 (0.0)	1 (12.5)
No cholesterol	10 (100.0)	6 (60.0)	4 (40.0)	0 (0.0)
No trans fat	46 (100.0)	41 (89.1)	2 (4.3)	2 (4.3)
Not fried	19 (100.0)	14 (73.7)	7 (36.8)	9 (47.4)

^1^ Data are presented as number (percentage). ^2^ High sugar: foods whereby sugar represented >10% of product calories. ^3^ High fat: foods whereby fat represented >30% of product calories. ^4^ High SFA: foods whereby SFA represented >10% of product calories. SFA: saturated fatty acid.

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
