# Peer review of "Marketing Strategy, Serving Size, and Nutrition Information of Popular Children’s Food Packages in Taiwan"

_nutrients, 2019, doi:10.3390/nu11010174_

Reviewer 1 Report

The introduction section is a bit misleading and it is difficult to follow. There seems to be a focus on supermarket with regard to child-targeted marketing strategies. Why is that so and why not considering other medias such as tv, internet and others. Also what is the age group targeted by the research: all children from birth to adolescence?

In materials and methods, criteria for the selection of foods included by the study are not clearly stated. With regard to nutritional information, it would have been important to mention why WHO`s draft recommendations did not complement that from Taiwan (Draft guidelines on saturated fatty acids... for adults and children). Similar comment for sodium:  (please, check guidelines for sodium intake in adults and children). Also, why using 240 ml as the reference for the serving. 

In the results section, it is surprising to see that many food products included in the study were not child-targeted while it was supposed to be the focus. In Table 3, it was expected that portions below 240 ml would have more likely less calories and sugar as compared to foods with portion sizes above 240 ml. 

The second and third paragraphs (lines 242-271) of the discussion section is a list of comparisons between results from the actual research and others. It would have been useful to explain similarities and difference between sets of results (actual and other studies). 

Reviewer 2 Report

The manuscript entitled “Association Among Child-Targeted Marketing, Serving Size and Nutritional Quality of Popular Children’s Foods in Taiwan” presents some interesting issues, but it requires some major corrections. 

 Title: 

I am not convinced that the title reflect fully the content. It is not “whole” marketing, but rather “label information on the package” being part of marketing strategy (but some of this information is regulated by the law)

 Abstract:

-Line 13 – “Large serving size increases energy intake and thus obesity risk” it should be “Large serving size may increase energy intake and thus obesity risk”

-Lines 13-14 – It is not true, as much attention has been given so far to the subject in literature and in public life. 

-Lines 18-19 – “ Compared with >240-mL/serving drinks, ≤240-mL/serving drinks had significantly fewer calories and less sugar” – it is obvious, that larger serving of the same product is characterized by a higher energy value.

-Term “health and nutrition marketing” should be specified 

-Lines 24-25 – “Ensuring that young children consume only small serving sizes of snacks and drinks may reduce childhood obesity and excessive energy and sugar consumption” Must be removed to not mislead readers. Preventing obesity in a group of children is not so easy! 

-Line 26 – What dis authors mean by “nutritional quality”?

 Introduction: 

-Lines 31-32 – It is rather supposition than correlation. In the original reference authors stated “Given that portion size trends coincided with the increasing prevalence of obesity in both the United States and Europe, it has been speculated that they are causally connected. Although these observational data cannot establish causality, they highlight the complexity of establishing a direct causal link between portion size and obesity, given that energy intakes are a function of not only the portion size of food, but also its energy density and the frequency of food and beverage consumption, among other factors”. Presented in this manuscript sentences and assumptions must be corrected and soften.  Correlation does not imply causation.

-Lines 36-36 – the reference 6 does not state exactly such things as suggested by authors of this manuscript. The main conclusion of this publication concerned something else. Authors should rewrite this section maintaining restraint.

-Lines 42-43 – please add the country (for what county/ city/ region) this result is stated.

-Line 44 – term “health and nutrition marketing” should be specified

-Line 45 – term “nutritional claims” is regulated by EU – this information must be presented - what can and what can not be “nutritional claims” on the package. Please add the information about legal/ law regulations regarding nutritional claims in Taiwan.

-Line 50 – authors must be more precise. This reference (no 13) in table 1 presented specific information that could be incorporated into the manuscript instead of some general statements that do not provide proper information

-Lines 53-54 – “but have paid little attention to the nutritional content of serving sizes of the children’s foods.” It is not true. 

 Materials and Methods:

-Line 62 - reference no 16 is not available on the main domain, therefore authors should describe the data collection in details. 

-Lines 72-73 –  It should be indicated that the research was conducted with no human subject and all collected data were publicly available, therefore no approval from ethical commission was required.

-More information about sample selection is required. 

-More information about coding form is required. This is curial due to the fact, that a number of researchers made coding. 

-Table 1 – please add the level of “High fat, high SFA, high sugar or high sodium” following by references

-Lines 95-97 – “10% of the total calories” – please add information about the chosen level of kcal due to the fact, that children in different age have a different energy demand. What cut-off was taken? I was total calories for children (what age?) or products? Similar situation is in the case of fat and SFA. 

-Line 99 – It concerns babies and toddlers (1-3 years old). For older children higher level was presented in this reference “9–50 year olds would be 200 and 400 mg sodium per serving which, when related to Canada’s Food Guide, meet the AI level of 1500 mg for this age group.” What is the recommended level in Taiwan? The data should be recalculated. 

 Results and discussion:

-The sections should be corrected accordingly

 Minor comments:

-Table 2 – please re-organize this table – it is really difficult to follow the data  (especially range of serving size)

-Figure 2 – green color of the background of this figure is inappropriate  

 Author Response

Please see the file

Round  2

Reviewer 1 Report

The introduction section is still difficult to follow and needs to be reorganized. The methodology has been clarified. Yet, one may again wonder why Canadian recommendations have been used to classify snacks and drinks with regard to their sodium content. There is a mistake in the results section (line 235): according to Table 5, 2,4% of the drinks were high sodium not 5%. The discussion should again be reviewed so as to explain the results and not just do comparisons of current findings with other studies. As such, what are reasons underlying this situation in Taiwan. 

Reviewer 2 Report

The article entitled „Marketing Strategy, Serving Size and Nutrition Information of Popular Children’s Food Package in Taiwan” has been improved and authors made a great effort to improve its quality. However I have still some comments:

 Abstract:

-Line 22 – “ More than 90% of product with sugar related claim were high sugar.” This result is alarming. If products has nutrition claim related to the sugar content (low in sugar) and have high level indeed, this is a fraud. How is it regulated by the government or control institution? What about the level of public trust? 

 Introduction:

-Line 41 “When children are served food in large serving sizes may lead to increased energy intake” please add “ but the energy density of serving sizes is important” 

 Materials and Methods:

-Line 113: „High fat: foods whereby fat content represented >30% of calories.” Please specify if it is 30% of calories from product or total calories from diet (similar lines, 114-115)

-Line 131; line 116, line 242 – This information are inconsistent (“High sodium: foods whereby the sodium content represented >260 mg sodium per serving” vs “Therefore, foods with 200 mg sodium per serving were defined as having high sodium content” vs “High sodium: foods whereby the sodium content represented >200 mg sodium per serving” ). It is a crucial issue due to the fact, that table 1 contains coding! 

-Table 1 – please verify if all “health claims” are “health claims” regulated by the law (please add references). I am not familiar with health claim “improve body defecation” – it must be some misunderstanding during translation. 

-Table 3 – The presented values (serving sizes and calories) are presented as mean values (±SD). Was the normality of distribution tested? It seems to be non-parametric distribution, but no median was presented. Authors must check it and the information about it should be added. If data have normal distribution should be treated as such (and Pearson correlation should be applied), if not, nonparametric tests should be applied (authors should use Spearman's rank correlation coefficient). 

-Table 3 – please add the calories per gram (density) 

-Table 4 – Authors applied non-parametric test. Pleased add what type of test authors used to verify the normality of distribution and if all values have non-parametric distribution

 Conclusion:

-The conclusions presented are too general and not related to the findings (e.g. “Parents need to pay attention to compare the calorie value and serving size among different food categories “)

-Authors should emphasize the findings and present the conclusions related only to the results. 

Author Response

Round  3

Reviewer 1 Report

I suggest to review the English. There are a few minor comments attached to the paper that may deserve further clarifications. 

Reviewer 2 Report

Manuscript entitled “Marketing Strategy, Serving Size and Nutrition Information of Popular Children’s Food Package in Taiwan” has been improved, but some corrections are still needed.

-Please add the sentence (from lines 141-142) “In Taiwan, there is no regulation about health claim.” – this sentence explained why the results are so alarming.

-Line 161 – It should be “Spearman's rank correlation” instead of “Spearman correlation”

-Line 166 – It should be “p of ≤0.05” instead “p of < 0.05” (see also - lines 217;236)
